# Sensory Modality in Students Enrolled in a Specialized Training Program for Security Forces and Its Impact on Karate Performance Indicators

**DOI:** 10.3390/jfmk10020114

**Published:** 2025-03-28

**Authors:** Ivan Uher, Ján Pivovarník, Mária Majherová

**Affiliations:** 1Institute of Physical Education and Sport, Pavol Jozef Šafárik University, 04180 Kosice, Slovakia; 2Faculty of Sports, University of Presov, 08001 Prešov, Slovakia; jan.pivovarnik@unipo.sk; 3Faculty of Humanities and Natural Sciences, University of Prešov, 08001 Prešov, Slovakia; maria.majherova@unipo.sk

**Keywords:** information transfer, karate training, perception, sensory modalities, thought experiment, learning

## Abstract

**Objectives:** The present study examined the sensory preferences adopted by students over three years of training in a specialized training program for security forces (STPSF). It determines their impact on karate performance metrics. **Methods:** Thirty-one students aged 20 to 26 (SD = 0.81) completed the modified Visual, Aural, Read/Write, and Kinesthetic questionnaire (VARK), a tool designed to help identify students’ preferred learning styles. This research suggests a theoretical model in which the balanced and optimal engagement of visual, auditory, and kinesthetic modalities rather than a strict mathematical equation might provide an optimal foundation for improving proficiency in martial arts. Balanced engagement of these sensory modalities can foster a deeper understanding of karate techniques, improve performance, minimize dependence on a single sensory channel, and bolster real-time adaptability. The students were tested at two points: once at the beginning of their enrolment and again after completing their three-year training program. **Results:** After a relatively intensive intervention over three years, the findings suggest a positive shift in the ratio of the primary modalities, moving toward an optimal balance. Considering the ideal sensory balance of 50:50:50%, the visual modality increased from 45.8 to 50.4, approaching the optimal value. The auditory modality, initially above the ideal level at 53.8, adjusted closer to balance, reaching 51.9. In contrast, the kinesthetic modality slightly decreased from 50 to 47.5, indicating a minor deviation from the ideal state. It was further confirmed that a higher technical level, such as the third kyu, exhibits an equal distribution, approaching the optimal use of the three modalities: visual 51.5 auditory 47.6 and kinesthetic 50.7. Moreover, the progress toward an optimal synergy and a more efficient evaluation of situational possibilities within the decision-making process was more frequently noted in females than in male students. **Conclusions:** Acknowledging students’ sensory processing preferences can assist the teacher, trainer, coach, and student in advancing interaction, optimizing learning strategies, improving performance, promoting analytical skills, and fostering self-assurance and determination.

## 1. Introduction

The integration of multiple sensory modalities in sports training is well established [1,2]. However, there is a growing interest in the application of multi-sensory learning, which incorporates visual, auditory, and kinesthetic modalities to enhance skill acquisition and performance [2]. This approach can be considered novel, as it explicitly integrates sensory learning theories to optimize training outcomes [1]. Some researchers and coaches increasingly recognize the importance of sensory input and cognitive processing in refining motor skills, improving situational awareness, and enhancing athletes’ overall performance [3,4]. This implies that identification of sensory input (visual, auditory, kinesthetic) and cognitive processing are crucial to optimizing skill execution, enhancing situational awareness, and improving performance. Real-time sensory signals enhance immediate corrections, while cognition supports decision-making and anticipation [5]. Balanced sensory integration optimizes reaction time, coordination, and skill execution, highlighting the importance of multi-sensory training in athletic development [6]. Athletes’ awareness of situational changes in play is unique, and is shaped by a variety of factors, both intrinsic and extrinsic, which are based on a complex interaction of one’s experience, learning styles, psychological state, physical condition, environment, and immediate context of sport, as claimed by Stone in 2012 [7]. These unique factors make every athlete’s interpretation of the same situation unique. This perspective has been widely discussed, as stated by Shuai, 2023 [8]. Research points to the practical linking of cognitive functions and motor learning with the optimization of input modalities; a theory that was proposed by Dosher, Ivanov, and Liu in 2024 [9,10,11]. According to Larcombe, 2017 and Green, 2015 [12,13] mental abilities such as attention, memory, and decision-making significantly influence how athletes learn and perform motor skills. In 2012, Schunk [3] stated that every sensory experience in sports leaves an imprint in the brain, facilitating neuroplasticity, which allows the brain to adapt and reorganize by forming new neural connections in response to learning and experience. This process enhances motor skills, reaction times, and decision-making in athletes. The brain strengthens movement, coordination, and sensory processing pathways, making actions more efficient and automatic, as Leysen asserted in 2011 [14]. Multi-sensory learning, engaging visual, auditory, and kinesthetic inputs, accelerates this adaptation, improving skill acquisition and performance. Furthermore, novel, challenging, emotionally charged, and multi-sensory experiences tend to create extensive neural imprints, while familiar, automatic actions or low-engagement tasks may lead to minute changes. Through consistent, deliberate practice and engagement, athletes can continuously reshape their brains to optimize performance. This perspective was formulated by Cho, 2011 and Keil, 2019 [15,16]. Dias, 2024 [17] asserts that the human brain learns new skills by interacting past experiences with new sensory information, using both present-moment focus and past knowledge as needed. Present-moment focus dominates when the task is novel or when fine-tuning is necessary, while past experiences guide automatic actions and decision-making in familiar situations [18]. Minifra [19] claims that vision and kinesthetic awareness are often crucial in present-moment experiences, where the balance of senses depends on the context of the activity or sport. Balancing all sensory inputs can enhance performance, but certain senses may take priority depending on the task [7]. However, when all senses (visual, auditory, and kinesthetic) are fully engaged and balanced, this typically means that the athlete is immersed in the task at hand, paying attention to real-time information from the subjects’ environment. That state of heightened sensory awareness reflects a strong present-moment focus, as the brain processes and integrates multiple sources of input to guide movement and decision-making [20]. Athletes are fully attuned to constant sensory feedback from their own bodies and cues from the environment. The balanced use of all senses enhances the mind–body connection, helping athletes adjust to present action based on immediate sensory input. Balanced sensory awareness means the brain operates in the now, not distracted by irrelevant thoughts [21]. Karate is a complex martial art that demands speed, precision, agility, and strategic decision-making. Mastering strikes, blocks, and footwork requires physical strength, spatial awareness, and reaction time. These skills are deeply influenced by sensory integration. Balancing visual input in karate (seeing the opponent’s movement), kinesthetic feedback (feeling body movement), and auditory cues (hearing the opponent’s own footsteps or breathing) ensures that no single sense is overburdened, allowing the brain to process information more efficiently. Thus, the key is to ensure that no single sense is overburdened and that the brain can integrate sensory input cohesively [22].

Based on our thought experiment, we propose a theoretical model suggesting that the equal engagement of visual, auditory, and kinesthetic modalities (50% each) leads to optimal sports performance. While we acknowledge that, mathematically, summing these percentages results in 150%, our approach does not imply an additive relationship but rather a balanced and proportionate utilization of sensory inputs. The 50:50:50 model does not mean that an athlete must designate a predetermined rate of attention to each modality in a bounded manner. Instead, it points out that each sensory channel should be activated at an adequate level for optimal performance, ensuring no modality is overly dominant or neglected. The percentage values should represent a model-based balance, where visual, auditory, and kinesthetic inputs function harmoniously to support athletic performance. Athletes who achieve this balanced sensory involvement may exhibit greater adaptability, precise efficiency in executing complex motor skills and maximizing biological efficiency. In the presented questionnaire, the reference to 50% per modality reflects our hypothesis that the most effective learning and performance occurs when all three sensory modalities are sufficiently and equally engaged. It does not imply an arithmetic total but rather represents a state of optimal sensory integration where no modality is underutilized or disproportionately dominant. Though the balance is conceptually proper, in practical application minor variations are inevitable due to rounding, individual differences, measurement limitations, and dynamic neuroplasticity. A 45–55% range per modality could be a more realistic approximation when discussing broader practical implementation. However, a smaller extent, 48–52%, of sensory engagement is possible and may be more realistic for individuals who naturally balance their sensory modality [23]. Balanced sensory engagement enhances learning, reduces cognitive overload, promotes present moment focus, supports flow or zone and sub-zone states, and increases versatility in real-time combat situations [24]. This balance efficiently allows for the processing of multi-sensory input, improving reaction time, decision-making, and overall reliance on one modality [25]. Therefore, the aim of this study was to examine the sensory modalities utilized by students throughout their three-year enrollment in a specialized training program for security forces (STPSF) and to evaluate their impact on karate performance. By assessing the distribution of visual, auditory, and kinesthetic inputs over time, we seek to determine how sensory engagement evolves during training and how it influences motor learning, skill acquisition, and overall performance in martial arts. A better understanding of these relationships will provide insights into the role of sensory integration in skill development, contributing to more effective training methodologies in karate and other combat sports. The study explores the sensory preferences adopted by students during three years of training in STPSF and assessed their influence on karate performance indicators.

## 2. Materials and Methods

### 2.1. Study Participants

Thirty-one students, 22 males and 9 females aged 20–26 (Table 1) were enrolled in the study. The modified VARK questionnaire was administered to students at two key points in their bachelor program for a specialized training program for security forces (STPSF) at a Slovak university. The first assessment occurred in September 2021, at the beginning of the first year of their three-year study program, during their initial enrollment. That allowed for a baseline measurement of their sensory preference before the commencement of specialized karate training. The second administration of the questionnaire occurred in June 2024, at the end of the third year, following the completion of the entire training program. The test was administered online to ensure that students could complete it at their own pace and convenience. Several measures were implemented to ensure the validity of the questionnaire administered at home. Detailed and consistent guidelines were given to ensure standardized interpretation. This standardized instrument VARK questionnaire was designed by Neil Fleming from New Zealand in 1987, was validated to ensure reliable data collection and was designed with neutral language. A time limit (15 min) was set to reduce external influences, and students were instructed to complete the questionnaire independently. An honesty statement was included to encourage truthful responses. The return of the tests was at 100% within one week, where the same standardized format could minimize potential external distractions or biases. Anonymity, comfort, and ease of data collection and storage enhanced the presented data handling. Using the same cohort of students for both assessments, the study aimed to track any shifts in sensory preferences that may have developed due to their intensive physical and aural training over the three years. Evaluations were restricted to students who completed the entire duration of the three-year program. Informed consent was secured from all participants before their engagement in the study. The students were fully informed about the study’s purpose and procedures, ensuring that their participation was voluntary, and they could withdraw without repercussions. Ethical approval for this case study was granted by the ethical committee of Prešov University for research. ID: ECUP022025PO.

### 2.2. The Modified Version of the VARK Questionnaire

A modified version of the VARK questionnaire was used to assess the sensory preferences of STPSF students. The VARK test is widely recognized and used in educational and training contexts to assess learning styles [26]. It provides a simple yet reliable framework for categorizing sensory preferences. The test typically evaluates four sensory modalities, visual, aural, read/write, and kinesthetic. Only visual, aural, and kinesthetic (VAK) questions were analyzed for the presented study. The VARK questionnaire comprised 16 questions, with equal representation across the four modalities. The decision to limit the scope to these modalities was based on the need to assess sensory preferences most relevant to physical and skill-based learning processes, such as those involved in karate training. For each sensory modality, the minimum possible score was 0, indicating no reliance on that modality. The maximum possible score was 16, representing the highest reliance on that modality across the entire questionnaire. If the answer did not match the student’s perception, a student could circle more than one answer or leave any question that does not apply blank. Following the completion of the questionnaire, a score chart was applied to measure their responses. Each questionnaire item was assigned a value, and the cumulative scores were calculated to reflect the participants’ overall preferences. A higher score in a particular modality reflected a stronger preference or reliance on that sensory modality for learning and skill acquisition.

A thought experiment was incorporated into this study to supplement the analysis of sensory preferences. This research assumed that a well-balanced sensory integration would entail an even distribution of reliance among the three modalities (visual, auditory, and kinesthetic). A ratio of 50:50:50% applied three times represents a hypothetical ideal sensory balance, allowing for the exploration of whether such equal stimulation correlates with improved karate performance. While the actual distribution of sensory preferences may vary among individuals, this theoretical model serves as a baseline to compare observed results and to investigate potential patterns in sensory dominance. As illustrated in Figure 1, we consider the possibility that the decision-making core within the brain contains 150 material carriers for all of the information supplied. It was assumed that these carriers can be occupied by different modalities (visual, auditory, and perceptual-kinesthetic), and our current decision-making depends on their occupancy. The core can be innovated in a certain way with new information, but only as a replacement for information already downloaded from the core. That is, by replacing it. The assumption is that the ratio of visual information carriers, auditory and perceptual (tactile–kinesthetic) in the core is 50:50:50% [27]. Where an optimal current decision-making process occurs within the body’s response to a stimulus, each of the modalities can thus optimally influence it. Scores from the modified VARK questionnaire were evaluated about this theoretical model to observe deviations and trends in sensory dependence. This approach establishes a baseline for analyzing results and considering balance in karate learning.

### 2.3. Outcome Variables

The primary outcome variables in the study were the scores for visual, auditory, and kinesthetic sensory modalities derived from the responses to the modified VARK questionnaire. These scores were calculated by summing the responses associated with each modality across the 16 questionnaire items. The score for each modality ranged from 0 (indicating no reliance on the modality) to 16 (indicating the highest reliance). These variables were used to identify participants’ dominant sensory modality and to analyze trends and variations in sensory preferences across the students. The total number of responses for each modality was counted and converted into percentages for each student. These percentages were then averaged across all participants to determine the overall sensory profile of the 1st and 3rd Y of the study group. The calculated percentages for each modality were compared with the ideal sensory balance of 50% visual, 50% auditory, and 50% kinesthetic proposed in the 50:50:50% thought experiment. The evaluation process also included assessing the technical proficiency level (TLP) set by the Slovak Karate Association.

### 2.4. Data Analysis

The sensory modality percentages were analyzed using descriptive statistics and parametric paired *t*-tests to compare individual modalities’ input and output measurements. The Kruskal–Wallis test was selected to compare the groups according to their technical proficiency level. Cohen’s d was used to evaluate the effect size and determine the practical relevance of the observed differences. Statistical significance was assessed at a level of (*p* < 0.05).

## 3. Results

When comparing students in the first year of study and the third year of exit in the three basic modalities—visual, auditory, and kinesthetic—no statistically significant changes were observed, though data trends showed notable shifts. Considering the ideal value in the ratio 50:50:50%, and as per Table 2, the visual modality approached the ideal 50.4 from 45.8, and the auditory was above the norm at 53.8 and approached 51.9. In the kinesthetic modality, there was a decrease from the ideal state of 50 by 2.5 points to 47.5. The relatively small sample size (n = 31) may have influenced the ability to detect statistically significant effects. While differences were observed in the visual modality, the lack of statistical significance could be due to insufficient statistical power rather than the absence of an actual effect.

Based on the findings, a histogram of the kinesthetic modality (Figure 2) was created, which shows how the structure has changed and shifted towards improvement.

The analysis of proficiency levels within the Kyu Grading System (fifth, fourth, and third kyu) demonstrated a progression toward higher ranks, with the most favorable outcomes observed at the third kyu, while the requirement is to achieve at least a fourth kyu upon graduation, see Table 3. In the visual modality, LTP values remained stable across ranks, indicating consistent proficiency development. Auditory processing exhibited evidence of adaptation and optimization, with TLP values reflecting a progressive optimization with rank advancement. The most substantial differences were observed in the kinesthetic modality, where TLP values indicated a pronounced progression, supporting an increasing reliance on kinesthetic processing at higher proficiency levels. Overall, the highest TLP values emerged at the third kyu, suggesting an optimal level of sensory integration and skill acquisition at this stage.

## 4. Discussion

The primary objective of this study was to examine which sensory modality (visual, auditory, or kinesthetic) students used when learning new skills and how that affects their performance in karate. The study employed a modified VARS questionnaire to assess student’s sensory preferences at the beginning and end of their three-year specialized training program for security forces (STPSF). The presented research aimed to determine if students’ learning styles change over time and whether optimal balance of 50:50:50% sensory engagement influence student’s skill acquisition and performance.

The data show that visual engagement increased from 45.8 to 50.4, moving toward the proposed optimal level, while auditory engagement decreased from 53.8 to 51.9, aligning more closely with balance. The kinesthetic modality declined slightly from 50 to 47.5, deviating from the ideal balance. These trends suggest a shift toward sensory equilibrium, partially supporting our hypothesis that prolonged training may promote a more balanced sensory involvement in karate performance. While notable differences in visual engagement were observed, statistical significance was not reached, possibly due to the small sample size and limited statistical power. However, the general trend toward more balanced sensory engagement suggests that multi-sensory integration plays a role in learning and skill refinement in karate. The results indicate a reciprocal adjustment between auditory and visual modalities, where one decreases while the other increases. This aligns with theories of multi-sensory learning [28], suggesting that, when one modality is excessively utilized, others may adapt to compensate. The kinesthetic decline could indicate a shift in sensory reliance based on task demands, warranting further exploration. Furthermore, analyzing the attained ranks within the Kyu Grading System (fifth, fourth, and third kyu) indicates a progressive improvement toward the third kyu. Notably, the visual modality exhibited the most stable results, with individual proficiency levels (TLP) showing no significant differences across ranks. In contrast, auditory processing demonstrated a pattern of adaptation and optimization through TLP, suggesting its role in refining skill acquisition over time. The most pronounced differences were observed in the kinesthetic modality, with a clear trend of improvement and refinement at the third kyu. This finding suggests that kinesthetic learning plays a crucial role in advancing skill proficiency. The results indicate that TLP is most favorably developed at the third kyu, reflecting this stage’s idealization of motor learning and sensory integration. The findings support the hypothesis that prolonged karate training influences sensory modality preferences, suggesting that these preferences are adaptable over time [28]. The presented findings highlight the importance of adaptive training methods to incorporate multiple sensory inputs to maximize skill acquisition in martial arts. It should be emphasized that there is a difference between memorizing and creative thinking in karate learning, where the distinction lies in the process and outcome of each. Memorizing is about repetition and accurate recall of existing information [27]. Creative thinking, on the contrary, is about synthesis and novelty. It involves thinking beyond the given information, making new connections, and imagining possibilities [29]. It is important to note that sensations lie within three primary classifications (interoception, proprioception, and exteroception), where combined sensations provide the student with essential information about themselves and the environment. The inability to process and integrate sensory input from the body and environment created overload and misperception, which is essential for students’ ability to engage in functional, meaningful, adaptive behaviors and learning.

The theory behind the 50:50:50 sensory balance in processing sensory information for sports and karate tasks is grounded in concepts of multi-sensory integration, optimal learning models, and motor control behavior [30,31,32,33]. The presented study provided support for this claim. A balanced state without a single sense of being overburdened allowed the student to make faster, more accurate decisions and execute movements effectively. Proprioceptive training (i.e., karate training) creates an effective strategy to achieve this, something which has also been confirmed by a study by Yilmaz and associates [34]. Here, the training minimizes students’ over-reliance on any single sensory channel, mitigates cognitive and physical strain, and facilitates more significant synaptic development and enhanced memory retention. It is a question of experiencing overall learning and perception in karate as a way of concentrating inside and outside the individual’s body [5,6,35,36]. Taken together, a dynamically balanced sensory processing system (50:50:50) across visual, auditory, and kinesthetic channels is critical for effective task execution. Any transient predominance of a single modality should be promptly corrected to sustain sensory coherence and prevent perceptual suppression, where an evenly distributed 50:50:50 sensory modality ratio ensures efficient physiological integration, leading to maximized energy efficiency. This can support neural optimization and foster the potential of achieving a flow-zone state. It can be stated that the students progressively enhanced their understanding and skill acquisition in karate.

When memorizing karate forms (kata), sensory perception primarily helps recall movements. Visual, auditory, and kinesthetic senses are engaged to store and retrieve precise patterns of techniques. The focus is on reproducing exact movement, repetition, and perfecting the pre-defined steps. While in creative learning, sensory perception goes beyond replication. It involves using the body’s feedback (from visual, auditory, and kinesthetic senses) to adapt and improvise within or outside the boundaries of prearranged forms. While the memorization of karate forms depends on sensory perception form accuracy and repetition, creative learning uses sensory perception more dynamically and proactively [6], allowing flexibility and improvisation to facilitate an equal distribution of sensory stimulation, 50:50:50, when designed and implemented effectively. It can be said that, while the enhancement in students’ creative thinking was modest, it was still evident. It is important to note that the memorization of karate forms may take less time initially, especially when the goal is to repeat the pattern or sequence, it may be easier to dissipate or forget over time than creative learning, as memorization is a more passive process, especially when without regular practice. The form might be retained in the short term, but the knowledge can weaken without deeper understanding or frequent reinforcement. Conversely, creative learning using 50:50:50 sensory engagement enhances understanding of the principles behind the movements and how they adapted to various scenarios, situations, and sensations, which generally takes longer to develop as it requires deeper engagement [6]. That leads toward longer-lasting retention, as it involves a more active, conceptual understanding. Once students understand the principles behind the movements, it is easier to reconstruct or adapt them, even after time away from practice. Moreover, creative learning can be a precursor to achieving a state of flow zone in performance [37,38,39]. Where creative learning fosters adaptability and flexible mastery, enhances focus, and encourages improvisation, experimentation, and innovation, which can lead to heightened emotional involvement (positive and subtle, calm focus), confidence, and autonomy, hence allowing the subject to perform effortlessly in a flow or zone state [40].

To achieve a more profound understanding and creative engagement, students must employ an optimal sensory interpretation framework that balances all three primary sensory modalities, where the visual modality provides clarity and understanding of spatial and structural relationships; the auditory modality adds depth through rhythm, verbal explanation, and sound-based cues; and the kinesthetic modality grounds understanding in physical experience, allowing the student to engage with the performance of the skill. By balancing these modalities, students avoid over-reliance on any one sensory channel, which can lead to impaired motor skills, reduced learning efficiency, poor decision-making, emotional and cognitive challenges, performance inconsistencies, and overcompensation that leads to mental and physical fatigue [41,42,43]. This process of sensory integration forms a feedback mechanism through which students absorb knowledge and elevate their skills and self-confidence. Maintaining a balance of sensory inputs fosters a dynamic and flexible understanding. By applying 50:50:50 sensory balance, students maximize their potential to adapt, improve, and innovate, achieving creative mastery rather than static non-dynamic skill acquisition.

The presented evidence supports the assertion that, when STPSF students are learning a new skill, they are often focused on acquiring the basics as quickly as possible. They may feel more comfortable relying on memorization and may fear making mistakes or being judged. Memorization is often a faster way for students to learn, especially under time constraints, like those associated with exams or deadlines [44]. Though memorization does not necessarily limit sensory adaptation in karate learning, over-reliance on memorization can hinder adaptability. Thus, memorization must be complemented with sensory engagement [45,46]. Furthermore, classroom learning in the Slovak education system emphasizes memorization without encouraging exploration or experimentation [47], therefore, students may default to memorization. Consequently, when students are conditioned to rely on memorization, they develop rigid learning habits (repeated practice enhances neuroplasticity by strengthening motor-neural pathways) whose inflexibility makes it harder for them to adapt to tasks that require creative problem solving. Student’s participants spend most of the time memorizing with restricted practice time (karate class), inferring that their creativity skills may remain sub-optimal. This in turn might restrict progress in karate, wherein STPSF students switch between a classroom that stresses correct answers and a karate class that priorities experimentation, convergent thinking and adaptation and which may create a cognitive dissonance. Students may struggle to reconcile the two approaches (memorization and creative learning), viewing them as unrelated rather than complementary, where the compartmentalization of experiences and reduced practice time can be recognized as factors limiting students’ advancement in presented examination.

However, despite the significant time students dedicated to memorization in the classroom, the findings revealed a positive shift in the ratio of primary sensory modalities, progressively moving toward a more optimal balance. Students who achieved higher technical levels demonstrated a more evenly distributed reliance on the sensory modalities, indicating an approach toward the ideal integration of visual, aural, and kinesthetic inputs. These results suggest that, with skill advancement, students naturally develop a more balanced and adaptive sensory engagement, even in an environment where memorization dominates. Nonetheless, investigating the role of sensory balance in learning and performance is an innovative and overlooked area [48,49,50], making the presented study more meaningful.

However, the outlined conclusions should be interpreted cautiously due to the limited sample size, which may impact statistical power and generalization. The sample, limited to STPSF students, may not apply to the broader population. Reliance on self-reported data may also introduce biases. The complexity of sensory balance measurement may only partially capture the dynamic interaction of modalities during karate performance. The study’s limitations could also include a deficiency of standardized external conditions and restricted evaluation criteria. However, these limitations enhance the study’s relevance by pointing to possibilities for further exploration and advancement of this meaningful contribution. The study also possesses several strengths, making the results particularly valuable for trainers, coaches, educators, and researchers in fields emphasizing physical skill acquisition, such as martial arts, security forces training, and sports science. The study’s strengths might include the its duration; tracking the same students over three years provides more reliable insights into changes in sensory learning preferences. By excluding read/write related questions, the study ensures a more accurate assessment of sensory preferences in karate training. This study contributes to a better understanding of how sensory modalities influence learning. Connecting learning styles with karate performance enhances the study’s practical relevance for skill development. Combining theory and practice to bridge the gap between the two modalities can also strengthen the presented study.

## 5. Conclusions

The presented examination investigated the sensory modalities employed by students through their three-year study in an STPSF program and analyzed their correlation with karate performance. Enhanced understanding of the interconnection between creativity, movement, and sensory integration is the foundation for optimal skill acquisition and performance enhancement. Students can optimize their potential by promoting optimal sensory integration and balancing visual, auditory, and kinesthetic modalities, moving beyond memorization to attain excellence and creativity. The data suggest that, despite the predominant use of memorization techniques in the classroom, a notable improvement was observed in the ratio of primary sensory modalities, achieving a more balanced distribution and correspondingly higher technical levels in karate. Integrating sensory input with creative thinking is fundamental for excellence in karate performance. The emphasis on perceptual alignment offers a practical framework for educators, coaches, and trainers to design effective training programs that optimize performance and foster innovation in karate. To detect students with imbalanced sensory modalities. Further research should explore the application of comprehensive sensory engagement across different sports and its impact on performance; psychological variables such as motivation, focus, and confidence are recommended.

## Figures and Tables

**Figure 1 jfmk-10-00114-f001:**
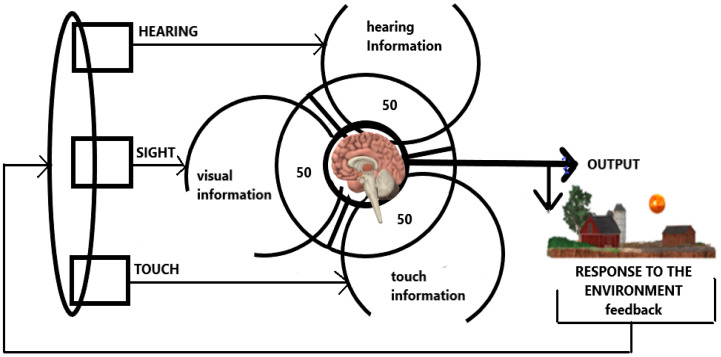
Based on the thought experiment, the 100 fundamental forms of core information are balanced equally across visual, auditory, and kinesthetic modalities 50:50:50.

**Figure 2 jfmk-10-00114-f002:**
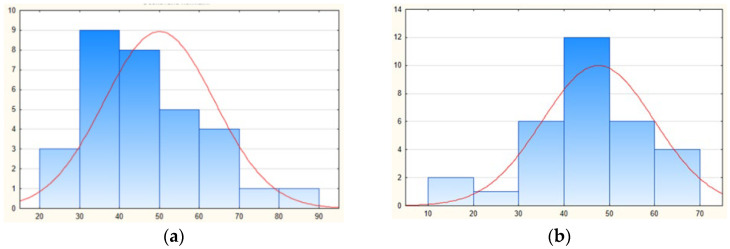
Histograms of the kinesthetic modality in the 1st (**a**) and 3rd grade (**b**) and its percentage changes in structure.

**Table 1 jfmk-10-00114-t001:** STPSF students’ age in years.

	N	min	max	Average	Median	Variance	SD
S (2021)	31	20	24	20.29	20	0.66	0.81
S (2024)	31	22	26	22.29	22	0.66	

Legend: S (2021)—students in the freshmen year of study, S (2024)—students in the senior year of study.

**Table 2 jfmk-10-00114-t002:** Comparison of modalities in the 1st and 3rd year of study.

Variable 1Y and 3Y		*t*-Test for Dependent Samples, Significance at *p* < 0.05							
	Mean	SD	n	Difference	STD	Cohen’s d	t	df	*p*
Visual Mod 1Y	45.81	18.08	31	−4.68	13.96	0.258	−1.87	30	0.072
Visual Mod 3Y	50.48	17.29							
Aural Mod 1Y	53.87	16.72	31	1.94	15.26	0.115	0.71	30	0.485
Aural Mod 3Y	51.94	16.52							
Kinesthetic Mod 1Y	50.00	13.84	31	2.42	15.27	0.157	0.88	30	0.385
Kinesthetic Mod 3Y	47.58	12.37							

Legend: Mod—modality, Y—year, SD—standard deviation, n—sample size, STD—standard deviation of differences, t—*t*-test, df—degree of freedom, *p*—significance level *p* < 0.05.

**Table 3 jfmk-10-00114-t003:** Modalities and their values according to the achieved TLP in 3 Y.

Students TLP	n	Visual Mo 3Y Mean	Visual Mo 3Y SD	Aural Mo 3Y Mean	Aural Mo 3Y SD	Kinesth. Mo 3Y Mean	Kinesth. Mo 3Y SD
kyu	31	51.54	17.72	47.69	16.66	50.77	14.84
4. kyu	31	50.00	17.97	53.93	17.12	46.07	10.22
5. kyu	31	48.75	17.97	58.75	13.77	42.50	10.41
TOG	31	50.48	17.29	51.94	16.52	47.58	12.37

Legend: TLP—technical level of proficiency, Kinesth.—kinesthetic, Mo—modality, TOG—two observed groups.

## Data Availability

The authors have full access to all specific material used in this paper and take responsibility for the use and accuracy of the information provided.

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
