# Peer review of "Sensory Modality in Students Enrolled in a Specialized Training Program for Security Forces and Its Impact on Karate Performance Indicators"

_jfmk, 2025, doi:10.3390/jfmk10020114_

Round 1
Reviewer 1 Report
Comments and Suggestions for Authors
Sensory Modality in Students Enrolled in a SpecializedTraining Program for Security Forces and Its Impact on Karate Performance Indicators
Ivan Uher, Ján Pivovarník and Mária Majherová
ABSTRACT In accordance with the topic, I suggested that the authors delete one key word.
INTRODUCTION has many shortcomings, which need to be corrected. The aim and hypothesis are not well defined. Also, the studies were not analyzed in detail, that is, the topic of the paper was not clearly analyzed and related to karate at all. Also, there are certain technical details that need to be corrected. The authors generally talked about sensory modality, cognitive functions, etc…. All that should be better connected with karate.
MATERIAL AND METHODS Overall this chapter is well written. However, certain details should be corrected. For example, the authors discussed the characteristics of the participants in the wrong chapter (Data Analysis).
RESULTS are clearly presented through tables and figures. Certainly, there are certain technical and professional details that need to be corrected. The authors also analyzed the Results and Discussion chapters within one chapter, which is wrong.
DISCUSSION is the chapter that was missing, and therefore many explanations are missing. I suggested that in this chapter the authors indicate the main results of the study. To explain and confirm the hypothesis, and compare their results with previous similar studies. To indicate whether the results were expected or not. On the other hand, it should be pointed out that the authors presented a lot of quality information about the STPSF program and correlations with performance in karate.
CONCLUSION In accordance with the obtained results.
REFERENCES are in accordance with the topic of the paper and listed according to the MDPI style. However, corrections are necessary, because some studies listed the authors' surnames in capital letters, which is unnecessary.
I SUGGEST MAJOR REVISIONS (26 comments).

Author Response
Line 34: thought experiment deleated from Keywords
Line 37: Karate mention, added according to the reviewer´s suggestion.
Line 38-40: references was added
[53] Lembo, L. Mariani, A.M. Maltisensory Training: Motor Learning and Sports Perfrormance in Young Athletes. J.of Inclusive Methodology and Technology in Learning and Teaching. 2021. Anno 1n.1.
[54] Lacalendola, Chiara, and Daniella Sinanagic. "The Power of the Five Senses: A Multisensory Brand Experience in Sport stores." 2020.
Line 46: Explain breafly that studies [2,3]
Some researchers and coaches increasingly recognize the importance of sensory input and cognitive processing in refining motor skills, improving situational awareness, and enhancing athletes' overall performance [2,3]. This implies that identification of sensory input (visual, auditory, kinesthetic) and cognitive processing are crucial to optimizing skill execution, enhancing situational awareness, and improving performance. Real-time sensory signals enhance immediate corrections, while cognition supports decision-making and anticipation [30]. Balanced sensory integration optimizes reaction time, coordination, and skill execution, highlighting the importance of multisensory training in athletic development [29].
Line 66: Modified - Minifra [14] claim
Line 71: Modified - athlete
Line 76: Modified - athletes
Line 82: Modified - athletes
Line 91: Explain 50:50:50 model
Based on our thought experiment, we propose a theoretical model suggesting that equal engagement of visual, auditory, and kinesthetic modalities (50% each) leads to optimal sports performance. While we acknowledge that mathematically, summing these percentages results in 150%, our approach does not imply an additive relationship but rather a balanced and proportionate utilization of sensory inputs. The 50:50:50 model does not mean that an athlete must designate a predetermined rate of attention to each modality in a bounded manner. Instead, it points out that each sensory channel should be activated at an adequate level for optimal performance, ensuring no modality is overly dominant or neglected. The percentage values should represent a model-based balance, where visual, auditory, and kinesthetic inputs function harmoniously to support athletic performance. Athletes who achieve this balanced sensory involvement may exhibit greater adaptability, precision, and efficiency in executing complex motor skills. In the presented questionnaire, the reference to 50% per modality reflects our hypothesis that the most effective learning and performance occurs when all three sensory modalities are sufficiently and equally engaged. It does not imply an arithmetic total but rather represents a state of optimal sensory integration where no modality is underutilized or disproportionately dominant.
Line 96: Modified - Therefore, the aim of this study was
Line 101: Modified - (Table 1)
Line 182, 196, 198: Modified - p < .05
Line 186: Modified - Discusssion deleted
Line 197, 214: Modified - Ledend
Line 202: Modified - Figure 1
Line 216: Added - Discussion
Line 228: Added - by Yilmaz and associates [28]
Line 241: Modified - optimization
Line 198: Discussion – was modified based on recommendations
Line 352: References – was modified

Reviewer 2 Report
Comments and Suggestions for Authors
Your paper presents an interesting and relevant topic, but there are areas that could benefit from improved clarity, logical flow, and stronger connections between different sections. Some parts of the discussion are somewhat disjointed, making it harder to follow the argument cohesively. Additionally, some claims would be more persuasive with additional numerical data and references. Consider refining transitions between ideas and reinforcing key points with supporting literature. Addressing potential limitations would also enhance the study’s credibility.
Lines 15-16: Consider using a period (0.81) instead.
Lines 17-19: The sentence "Included in the experimentation was a concept based on a thought experiment, which proposed that equal sensory distribution—50% each for visual, auditory, and kinesthetic inputs—might provide an optimal foundation for improving proficiency in martial arts." is somewhat unclear regarding the percentage distribution. Could you clarify it further? Ensuring clarity in the abstract would enhance readability and impact.
Lines 24-30: Since results are being discussed, numerical data should be included to provide more substance.
Line 37: The phrase "The multi-sensory learning approach in sports is not entirely novel" does not read smoothly. Consider rewording for better fluency.
Line 55: Neuroplasticity is mentioned but not sufficiently explained. Given its relevance to how athletes learn, a brief explanation would be beneficial. Additionally, adding more references would strengthen the discussion.
Lines 50-69: While your main argument is clear, the section lacks cohesion. The ideas are somewhat disjointed, even though they make sense individually. Improving the text’s flow would enhance readability.
Line 86: This is the first time you introduce this sport in the text. Consider adding a few sentences about its key characteristics and the skills it requires before tying it to your research objective. Additionally, the section would benefit from smoother transitions between ideas.
Lines 91-95: You propose an ideal sensory distribution model of 50:50:50, but there is no discussion of potential variation margins or experimental errors. It would be helpful to indicate an acceptable range if applicable.
Lines 92-98: The final discussion needs refinement to more effectively highlight your objective. The wording is currently somewhat rough.
Line 98: The acronym STPSF is already explained in the abstract. I suggest reversing the order—keep only the acronym in the abstract and explain it fully when it first appears in the introduction. Alternatively, if you prefer to define it in both sections, ensure it is written out in full in both instances.
Line 102: The same suggestion as in line 98 applies to this acronym.
Lines 108-110: The VARK questionnaire was administered online. To strengthen validity, specify whether any measures were taken to prevent distractions or external influences during completion.
Lines 157-162: The explanation of the decision-making core is compelling, but the concept of 150 material carriers requires either a bibliographic reference or an example for clarity.
Lines 188-189: "We did not observe statistically significant changes, but we noticed shifts in the data." is somewhat redundant. A more concise alternative could be: "No statistically significant changes were observed, but data trends showed notable shifts." If you agree, this phrasing may enhance clarity.
Lines 193-195: Could this be related to low statistical power? Have you calculated it?
Line 208: Although an acronym legend for TLP is provided in Table 3, it is first mentioned in the text before the table. Consider defining it directly within the text for clarity.
Lines 246-256: Several claims in this section require stronger support from the literature. The distinction between memorization and creative thinking is interesting, but the link between these concepts and your study’s data is not clearly articulated. I suggest clarifying how the data substantiate your research objective.
Line 268: It may be more appropriate to use the past tense: "they adapted."
Line 357: Have you considered discussing any limitations? Addressing them would enhance transparency and credibility.
I am confident that with the right modifications and careful refinements, this work will be ready for publication.
Comments on the Quality of English LanguageThe manuscript contains some grammatical errors and awkward phrasing that affect clarity and readability. Certain sentences would benefit from rewording to improve their flow and coherence. Additionally, some sections use complex or ambiguous structures that make comprehension more difficult. A thorough revision of the language, focusing on sentence structure and readability, would enhance the clarity and impact of the paper. Consider reviewing key sections to ensure precise and fluid expression of ideas.
Author Response
Line 15-16: 0,81 0.81
Line 17-19: equal sensory distribution -50% each for visual, auditory and kinesthetic inputs –might provide…. This study proposes a theoretical model in which the balanced and optimal engagement of visual, auditory, and kinesthetic modalities rather than a strict mathematical equation
Line 24-30: Numerical data should be included to provide more substance.
Considering the ideal sensory balance of 50:50:50%, the visual modality increased from 45.8 to 50.4, approaching the optimal value. The auditory modality, initially above the ideal level at 53.8, adjusted closer to balance, reaching 51.9. In contrast, the kinesthetic modality showed a slight decrease from 50 to 47.5, indicating a minor deviation from the ideal state. It was further confirmed that a higher technical level, such as the third kyu, exhibits an equal distribution, approaching the optimal use of the three modalities visual 51.5., auditory 47.6., and kinesthetic 50.7.
Line 37: The multi-sensory learning approach in sports is not entirely novel. The integration of multiple sensory modalities in sports traning is well-established
Line 56: Explain neuroplasticity (brief explanation).
Neuroplasticity is the brain´s ability to adapt and reorganize by forming new neural connections in response to learning and experience. This process enhances motor skills, reaction times, and decision-making in athletes. The brain strengthens movement, coordination, and sensory processing pathways through repetitive practice, making actions more efficient and automatic [48]. Multi-sensory learning, engaging visual, auditory, and kinesthetic inputs, accelerates this adaptation and improves skill acquisition and performance.
Line 50-69 Enhance readability. Research points to the practical linking of cognitive functions and motor learning with optimizing input modalities [5,6,7]. Several scholars [8,9] suggest that mental abilities such as attention, memory, and decision-making significantly influence how athletes learn and perform motor skills. The [2] states that every sensory experience in sports leaves an imprint in the brain, facilitating neuroplasticity, which allows the brain to adapt and reorganize by forming new neural connections in response to learning and experience. This process enhances motor skills, reaction times, and decision-making in athletes. The brain strengthens movement, coordination, and sensory processing pathways, making actions more efficient and automatic [48]. Multi-sensory learning, engaging visual, auditory, and kinesthetic inputs, accelerates this adaptation, improving skill acquisition and performance. Furthermore, novel, challenging, emotionally charged, and multi-sensory experiences tend to create extensive neural imprints, while familiar, automatic actions or low-engagement tasks may lead to minute changes. Through consistent, deliberate practice and engagement, athletes can continuously reshape their brains to optimize performance [11,12]. The [49] asserts that human brain learns new skills by interacting past experiences with new sensory information, using both present moment focus, and past knowledge as needed. Present-moment focus dominates when the task is novel or when fine-tuning is necessary, while past experiences guide automatic actions and decision-making in familiar situations [13]. Some researchers claim [14] that vision and kinesthetic awareness are often crucial in present-moment experiences. The balance of senses depends on the context of the activity or sport. Balancing all sensory inputs can enhance performance, but certain senses may take priority depending on the task [1].
Line 86: Adding a few key characteristics and the skills of karate.
Karate is a complex martial art that demands speed, precision, agility, and strategic decision-making. Mastering strikes, blocks, and footwork requires physical strength, spatial awareness, and reaction time. These skills are deeply influenced by sensory integration.
Line 91-95: discusses potential variation margins.
Even though the balance is conceptually proper, but in practical application, minor variations are inevitable due to rounding, individual differences, measurement limitations, and dynamic neuroplasticity. A 45-55% range per modality could be a more realistic approximation when discussing broader practical implementation. Yet, narrower range 48-52% sensory engagement is possible and may be more realistic for individuals who naturally balance their sensory modality [54].
Line 92-98: highlight more effectively your objective.
This study aims to examine the sensory modalities utilized by students throughout their three-year enrollment in the Specialized Training Program for Security Forces (STPSF) and to evaluate their impact on karate performance. By assessing the distribution of visual, auditory, and kinesthetic inputs over time, we seek to determine how sensory engagement evolves during training and how it influences motor learning, skill acquisition, and overall performance in martial arts. Understanding these relationships will provide insights into the role of sensory integration in skill development, contributing to more effective training methodologies in karate and other combat sports.
Line 108-110: How we strenghten validity of the questionnaire?
Several measures were implemented to ensure the validity of the questionnaire administered at home. Detailed and consistent guidelines were given to ensure uniforme understanding. This standardized instrument was validated to ensure reliable data collection and was designed with neutral language. A time limit (15minutes) was set to reduce external influences, and students were instructed to complete the questionnaire independently. An honest statement was included to encourage truthful responses.
Line 157-162: the concept of 50:50:50% requires a bibliographical reference.
Williams, A.M. Jackson, R. Anticipation and Decision Making in Sport. 2019. pub. Routledge. p.428.
Line 193-195: Statistical power.
The relatively small sample size (N=31) may have influenced the ability to detect statistically significant effects. While differences were observed in the visual modality, the lack of statistical significance could be due to insufficient statistical power rather than an absence of an actual effect.
Line 188-189: Revised according to the reviewer´s suggestion.
no statistically significant changes were observed, but data trends showed notable shifts.
Line 208: Revised according to the reviewer´s suggestion.
technical proficiency level (TLP)
Line 246-256: Include: distinction between memorization and creative thinking, data substantiation and support from literature.
The primary objective of this study was to examine which sensory modality (visual, auditory, or kinesthetic) students used when learning new skills and how that affects their performance in karate. Study employed a modified VARS questionnaire to assess student’s sensory preferences at the beginning and end of their three-year specialized training program for security forces (STPSF). Presented research aimed to determine if students learning style change over time and whether optimal balance of 50:50:50% sensory engagement influence student’s skill acquisition and performance. The data show that visual engagement increased from 45.8 to 50.4, moving toward the proposed optimal level, while auditory engagement decreased from 53.8 to 51.9, aligning more closely with balance. The kinesthetic modality declined slightly from 50 to 47.5, deviating from the ideal balance. These trends suggest a shift toward sensory equilibrium, partially supporting our hypothesis that prolonged training may promote a more balanced sensory involvement in karate performance. While notable differences in visual engagement were observed, statistical significance was not reached, possibly due to the small sample size and limited statistical power. However, the general trend toward more balanced sensory engagement suggests that multisensory integration plays a role in learning and skill refinement in karate. The results indicate a reciprocal adjustment between auditory and visual modalities, where one decreases while the other increases. That aligns with theories of multisensory learning [55], suggesting that when one modality is excessively utilized, others may adapt to compensate. The kinesthetic decline could indicate a shift in sensory reliance based on task demands, warranting further exploration. Furthermore, analyzing the attained ranks within the Kyu Grading System (5th, fourth, and third kyu) indicates a progressive improvement toward the third kyu. Notably, the visual modality exhibited the most stable results, with individual proficiency levels (TLP) showing no significant differences across ranks. In contrast, auditory processing demonstrated a pattern of adaptation and optimization through TLP, suggesting its role in refining skill acquisition over time. The most pronounced differences were observed in the kinesthetic modality, with a clear trend of improvement and refinement at the third kyu. This finding suggests that kinesthetic learning plays a crucial role in advancing skill proficiency. The results indicate that TLP is most favorably developed at the third kyu, reflecting this stage's idealization of motor learning and sensory integration. The findings support the hypothesis that prolonged karate training influences sensory modality preferences, suggesting that these preferences are adaptable over time [56].
Presented findings highlight the importance of adaptive training methods to incorporate multiple sensory inputs to maximize skill acquisition in martial arts. It should be emphasized that there is a difference between memorizing and creative thinking in karate learning, where the distinction lies in the process and outcome of each. Memorizing is about repetition and accurate recall of existing information [51]. Creative thinking, on the contrary, is about synthesis and novelty. It involves thinking beyond the given information, making new connections, and imagining possibilities [52].
Line 268: Revised according to the reviewer´s suggestion.
they adapted
Line 357: Have you considered discussing any limitations?
The study limitations of this research are discussed in lines 311-318

Round 2
Reviewer 1 Report
Comments and Suggestions for Authors
In general, the authors of the paper have mostly taken the reviewers' comments and suggestions into account. However, there are certain issues that the authors need to address. For example, the hypothesis in the introduction needs clarification, and in the results, the Cohen's d effect size values are missing, which would provide a better insight and clearer picture of the obtained results. Additionally, I must emphasize that the authors need to pay attention to the order of citation of references in MDPI, JFMK. This part regarding the references has not been properly done and requires changes both in the text and at the end in references.
I suggest Major Revisions 12 comments

Author Response
Comments to the Reviewer 1
Line 34: References are corrected
Line 76: Point deleted
Line 104: Included
The study explores the sensory preferences adopted by students during three years of training in STPSF and assesses their influence on karate performance indicators.
Line 125, 191, and 207: Modified in Tables 1,2,3
Lines 171 and 191: Cohen’s d added.
Line 187: n=31 modified
Line 207: Table 3. n – modified
Line 319: Modified first limitations, then strengths

Reviewer 2 Report
Comments and Suggestions for Authors
I would like to thank the authors for addressing my requests. Overall, the manuscript is well-structured, and the content is relevant. I only have a few minor corrections, which I have included in the comments. Additionally, I recommend conducting a final check to ensure the overall fluency of the text.
Lines 47-63: I suggest including the year when citing these studies to provide the reader with a temporal perspective on the topic’s evolution (e.g., “In 20xx, Schunk states…”).
Line 76: There are two periods; remove the one before the citation.
Line 105: “Thirty-one students, 22 males and nine females” be consistent in writing numbers, either in words or digits, but avoid alternating between the two.
Line 114: “A time limit (15 minutes) was set to…” use a consistent format for time units, either 15’ or min.
Line 251: After reviewing the figure, I suggest improving its clarity by slightly reducing the thickness of the lines and making it more visually organized.
Author Response
Comments to the reviewer 2
Line 46-55: Authors of the claims added.
Line 78: Period removed.
Line 109: number modified for consistency
Line 118: modified to (15 min.)
Line 251: The thickness of the lines was reduced.

Round 3
Reviewer 1 Report
Comments and Suggestions for Authors
The authors have made revisions in accordance with the reviewer’s suggestions, and I believe that the paper is now ready for publication in JFMK. The only thing the authors should do is provide a more detailed explanation of the Cohen’s d effect size values in Table 2 within the Results section.
Also, the last sentence of the introduction should state: "The research hypothesis was..."
Kind regards